# Hip Preservation Surgery in Osteoarthritis Prevention: Potential Benefits of the Radiographic Angular Correction

**DOI:** 10.3390/diagnostics12051128

**Published:** 2022-05-02

**Authors:** José M. Lamo-Espinosa, Adrián Alfonso, Elena Pascual, Jorge García-Ausín, Miguel Sánchez-Gordoa, Asier Blanco, Jorge Gómez-Álvarez, Mikel San-Julián

**Affiliations:** 1Orthopedic and Traumatology Surgery Department, Clínica Universidad de Navarra, 31008 Pamplona, Spain; adrianalfonsoe@gmail.com (A.A.); jgarciaausin@gmail.com (J.G.-A.); msanchezrui.1@alumni.unav.es (M.S.-G.); ablanco.7@alumni.unav.es (A.B.); jgomeza@unav.es (J.G.-Á.); msjulian@unav.es (M.S.-J.); 2Complejo Hospitalario de Navarra, 31008 Pamplona, Spain; elena.pascual.roquetjalmar@gmail.com

**Keywords:** hip preservation surgery, femoroacetabular impingement, hip arthroscopy, radiographic adult hip dysplasia

## Abstract

Objective: The aim of the study is to describe the morphology associated with the development of osteoarthritis (OA) in three different age groups. These data will contribute to defining the morphology associated with early and late hip OA. Methods: We studied 400 hips in 377 patients who had undergone primary THA due to idiopathic OA. Three groups were compared: group 1 (*n* = 147), younger patients, aged up to 60 years; group 2 (*n* = 155), patients aged between 61 and 74 years; and group 3 (*n* = 98), aged 75 or over. Five independent researchers measured the hip angles and the mean values were used to build a database. Results: No differences between groups in sex distribution and BMI were detected. Less coverage of the head (extrusion index), higher Tönnis angle, lower Wiberg and alpha angles characterized early OA hips. These differences increased with age, being greater between group 2 and group 3 (*p* < 0.01). However, significant differences were still present in the comparison between group 1 and group 2 (*p* < 0.01)). No differences were detected between group 2 and group 3. Conclusion: Elevated acetabular angle, head extrusion and decreased Wiberg angle characterize hip osteoarthritis at younger ages and should be the focus of hip preservation surgery in terms of osteoarthritis prevention. Pincer-type FAI (higher Wiberg and lower Tönnis angle) and higher alpha angle (CAM) are correlated with the development of later OA. These results shed doubt on applying the hip preservation surgery concept in terms of osteoarthritis prevention in FAI, especially in Pincer-type FAI patients.

## 1. Introduction

Hip preservation surgery (HPS) has grown in popularity in the last three decades. The procedures associated with HPS are focused on the correction of the anatomic morphology that has been claimed to lead to osteoarthritis (OA) in order to avoid hip arthroplasty. This mechanical paradigm has been postulated for many years [1,2,3,4,5,6]. Two major entities have been identified as a major cause of hip OA: femoroacetabular impingement (FAI) and adult hip dysplasia (AHD) [5].

Although some previous descriptions existed previously, the concept of FAI was coined by Myers et al. in 1999, and was suggested by Ganz as being the underlying cause of early idiopathic hip OA in 2003. There are three types of FAI. CAM-type FAI involves a hump in the transition area between the femoral head and the neck, so with flexion and rotation of the hip, this hump hits the labrum and the acetabulum. Pincer-type FAI is an overgrowth of the acetabular rim that causes friction against the head of the femur and its cartilage when flexing and rotating the hip. The third type is when CAM-type and Pincer-type deformities appear in the same patient [5]. (Figure 1) The presence of the CAM morphology in quadruped animals is the rule and, theoretically, activity and bipedal standing might have led to humans having a lower CAM prevalence [7,8,9,10]. In spite of this, some major studies have reported a prevalence of 42% to 75% for FAI x-ray signs in asymptomatic individuals [10,11,12,13,14,15]. These data raise doubts about what is and what is not normal.

Acetabular dysplasia is characterized by insufficient containment of the femoral head by the acetabulum, with the global femoral head under covered as a result of the insufficient bony containment. Uncorrected symptomatic acetabular dysplasia may increase the risk of functional impairment and degenerative joint disease due to distribution of joint reactive forces across a narrow segment of articular cartilage, associated labral hypertrophy and pathology, bony impingement and capsular attenuation [16,17,18,19]. Previous studies have linked acetabular dysplasia to premature osteoarthritis and the increased likelihood of requiring arthroplasty [20,21,22]. It has been estimated that acetabular dysplasia is responsible for 20% to 50% of Americans suffering from symptomatic hip osteoarthritis [23].

Our aim is to describe the morphology associated with the development of OA in three different age groups. These data will contribute to defining the morphology associated with early and late hip OA.

## 2. Materials and Methods

We report a cross-sectional descriptive study of 400 hips in 377 patients who had undergone primary THA in our institution. We included the first 400 primary THA secondary to idiopathic OA from our database, which includes patients since 2008 (the year when digital records were launched in our institution). We excluded THA secondary to other etiologies and those with no radiological studies in their medical history.

To carry out this study, three groups were evaluated.

-Group 1 (*n* = 147): Group corresponding to younger patients aged 60 or less;-Group 2 (*n* = 155): Group corresponding to patients aged between 61 and 74 years;-Group 3 (*n* = 98): Group corresponding to elderly patients aged 75 years or more.

We compared gender and BMI in the three groups in order to determine that the groups were comparable and that the only difference between them was age.

### Radiological Measurements

We used two views of the hip. The preoperative study included an anteroposterior and axial X-ray following the recommendations published by Campbell [24]. Five independent researchers measured the hips (Figure 2) and the mean value was used to build the research database:

The alpha angle is the angle formed by the axis of the femoral neck and a line connecting the center of the femoral head to the point where the asphericity of the head first is evident [25]. It is measured in the axial view. An alpha angle >55° defines CAM-type FAI.

The centrum–collum–diaphyseal (CCD) angle is the angle formed by the anatomical axis of the femur and the axis of the femoral neck. An angle <125° is known as Coxa vara.

The Tonnïs angle, or acetabular index, is formed by a line connecting the two ends of the acetabular sourcil and the horizontal line of the pelvis [26]. An angle <0° is suggestive of a Pincer-type FAI.

The lateral center-edge or Wiberg angle is the angle formed by a line drawn through the center of the femoral head and perpendicular to the horizontal line of the pelvis and a line connecting the center of the femoral head and the lateral lip of the acetabulum. Its normal values range from 25° to 40°. If the angle is less than 25°, we can classify this as hip dysplasia [27].

The femoral head extrusion index is defined as the percentage of the femoral head that is not covered by the acetabulum. An extrusion index > 25% is suggestive of hip dysplasia [28].

Sample size calculation: Approximately 32,000 hip arthroplasties are performed in Spain each year. A sample size of 382 patients was required to achieve 95% power, assuming a standard deviation for both groups of 10, and an alpha value of 0.05 (two-sided test).

Statistical analysis: Data were summarized using means and standard deviations (SD), medians and interquartile ranges and also percentages. The Kolmogorov–Smirnov test was used to test the normality assumption. For comparisons, we used the unpaired ANOVA, with post hoc tests as appropriate. All tests were two-tailed. A *p* value < 0.05 was considered statistically significant. All the statistical analyses were performed using SPSS 21 (Stata Corp. 2015. Stata Statistical Software: Release 14. College Station, TX, USA: Stata Corp LP).

## 3. Results

Demographic Data: The sample consisted of 400 hips in 377 patients. Demographic data are summarized in Table 1. The average age of the patients was 64.75 (SD:11.8) years. Three groups showed similar baseline characteristics regarding body mass index (BMI) and sex distribution (*p* > 0.05).

Radiological Analysis (Table 2 and Table 3): In early OA, the hips described have less coverage of the head, open Tönnis angles, lower alpha angles and closed Wiberg angles. These differences increase with the age, being bigger in the comparison between group 1 and group 3. Results are summarized in Table 2 and Table 3.

The Wiberg Angle was 30.54 (SD:14.63), 41,73 (SD:11.20) and 44.77 (SD: 12.02) for groups 1, 2 and 3, respectively. Wiberg angle differences were present between group 1 and group 2, with a mean difference of −11.39 (IC95%: (−15.034–(−7.75)) (*p* < 0.01), and between group 1 and group 3, with a mean difference of −14.43 (IC95%: −18.58–(−10.2)); (*p* < 0.01). No differences were detected between group 2 and group 3 (*p* = 0.133).

The Tönnis angle was 15.35 (SD:8.77), 8.94 (SD:8.63) and 7.11 (SD: 10.57) for groups 1, 2 and 3, respectively. Tönnis angle differences were present between group 1 and group 2 with a mean difference of 6.41 (IC95%: 4.00–9.8) (*p* < 0.01), and between group 1 and group 3 with a mean difference of 8.24 (IC95%: 5.12–11.36) (*p* < 0.01). No differences were detected between group 2 and group 3 (*p* = 0.395).

The extrusion index was 22.12 (SD: 4.54); 16.73 (SD: 11.47) and 13.89 (SD: 11.03) for group 1, 2 and 3 respectively. Extrusion index differences were present between group 1 and group 2, with a mean difference of 5.39 (IC95%: 2.12–8.66) (*p* < 0.01), and between group 1 and group 3, with a mean difference of 8.23 (IC95%: 4.61–11.85) (*p* < 0.01). No differences were detected between group 2 and group 3 (*p* = 0.65).

The alpha angle was 61.78 (12.87), 71.34 (17.15) and 73.64 (19.46) for groups 1, 2 and 3, respectively. Differences in the alpha angle were present between group 1 and group 2, with a mean difference of −9.55 (IC95%: −14.65–(−4.46)) (*p* < 0.01), and between group 1 and group 3, with a mean difference of −11.86 (IC95%: −18.00–(−5.71)) (*p* < 0.01). No differences were detected between group 2 and group 3 (*p* = 0.72).

The cervical–diaphyseal angle was 133.17 (8.96), 130.60 (8.61) and 130.28 (9.34) for groups 1, 2 and 3, respectively. No differences were detected between groups.

## 4. Discussion

Our research correlates the morphology of the hips with the timing of surgical treatment for hip osteoarthritis. These data could help the hip surgeon to consider the morphology that could lead to OA and the value of performing HPS at that level.

Higher Wiberg and lower Tonnïs angle and extrusion index have been associated with later OA development and correlate with Pincer-type morphology. One of the objectives of the Pincer-type FAI intervention is to increase the Tonnïs angle and decrease the Wiberg angle. Some authors have reported previously the protective effect of Pincer-type FAI against OA development because of a higher surface and a homogenous distribution of the load [6,27,29,30]. In a significant research paper, Nicholls et al. did not associate the greater Wiberg angle (Pincer-type FAI) with a greater probability of THR placement after 19 years of follow-up in a representative sample of a bigger cohort of 1003 subjects [6]. Acetabuloplasty reduces the load surface and requires a partial detachment and reattachment of the labrum, which could affect the vacuum effect that the hip possesses with a fully intact labrum. We should note that the presence of the pincer shape has been described in a high percentage of the asymptomatic population. Franck et al., in a study of 2114 hips of asymptomatic patients with a mean age of 25 years, observed the presence of the Pincer-type form in 67% of cases [12]. Therefore, it seems reasonable that, since the pincer effect does not seem to correlate with the development of early osteoarthritis, the orthopedic surgeon should at least be careful when indicating acetabuloplasty, especially in cases where the labrum is not damaged.

We did not find any association between the CAM-type FAI and the development of early OA. The patients with the highest alpha angle were those who later received surgical treatment. Hartofilakidis et al. reported that the alpha angle did not influence the development of osteoarthritis, being even higher in those cases in which OA did not develop over the 18-year follow-up [31]. In the same manner, Wyles et al. compared the 20-year evolution to OA of hips with and without FAI syndrome. Patients with CAM-type FAI had, at 20 years’ follow-up, the same probability of receiving an arthroplasty as those with normal alpha angle. Wyles concluded that HPS in FAI to prevent OA has a minimal impact on the possible evolution to osteoarthritis [32]. In addition, Barkados et al., in 2009, found evidence in patients with mild osteoarthritis and CAM-type FAI that osteoarthritis progression after 10 years’ follow-up was present in only one-third of cases, suggesting that there are other etiologic factors than might influence progression to OA. Unfortunately, no control group without CAM was evaluated [27].

The association between the development of OA and the CAM-type morphology has been previously described secondary to labrum injury [5,33]. However, there is no clear evidence that labrum lesions are more frequent in patients with CAM-type or Pincer-type FAI. The prevalence of labrum lesions has been reported as 80–96%, with no relationship between CAM morphology and ruptures [14,34,35,36,37]. Byers et al., in a study of 365 cadaveric hips, found that in subjects over 30 years of age, 88% of the hips exhibited labrum injury. Although the relationship between labral lesions and CAM was not studied, a high prevalence of labrum lesions seems to be the rule in people over 30 years old [35]. In another cadaveric study, Seldes et al. examined 55 cadaveric hips with a mean age of 78 years, finding that 96% of them presented labrum injury, of which 67% did not present OA changes [37]. Philippon et al., in a 2013 study of asymptomatic individuals, observed a prevalence of Cam-type FAI in 75% of ice hockey players and 40% of skiers. Curiously, despite this difference in the prevalence of the CAM morphology, which suggests an acquired etiology, no differences in labrum lesions were reported between groups [14]. More etiological factors must influence the progression of OA, because the mechanical labrum theory associated with FAI and labrum lesions is still unclear, and as a consequence, the association between FAI (both CAM- and Pincer-types) and hip OA development is not well explained.

There is no evidence of any correlation between CAM-type FAI and hip pain. In an important study of 3202 patients, Gosvig et al. found no correlation between CAM morphology and mechanical groin or hip pain. Accordingly, no correlation between CAM morphology and hip OA was reported [38]. It has been reported that 73.8% of the patients with unilateral hip pain exhibit CAM morphology in the other, asymptomatic hip [39]. Along similar lines, asymptomatic volunteers without a history of hip pain have been described as having CAM deformities in 35–75% of cases, usually present on both sides [40,41,42]. These data show the importance of the CAM shape and its low association with pain. We must remember that the most frequent situation in patients with a raised alpha angle is that they do not experience pain, and the correction of this angle to a normal value by an experienced hip arthroscopic surgeon is not well correlated with pain improvement [14,43]. In fact, the clinical results of osteochondroplasty associated with labral repair in several randomized clinical trials were not superior to a full program of physical therapy or arthroscopic saline lavage without performing any arthroscopic hip surgery [44,45,46].

The alpha angle was greater in older patients (group 3). Doubts arise about the origin of this morphology, in terms of whether it is congenital or acquired [39,47,48]. It could be secondary to the OA process, as a classic location of femoral osteophytes. In 2014, LaFrance observed that the prevalence of the CAM-type morphology in people over 65 years of age is 70.47% compared to 24.62% in those under 25 years of age, suggesting that the presence of the CAM-type form is acquired or secondary to the degenerative process [48]. In addition, the presence of chondropathy at the time of the intervention has been described in up to 72% of cases [47]. Haneda et al., in a recent paper, identified metabolic hyperactivity in cartilage from patients with early-stage FAI, suggesting that inflammation and degeneration markers are the same as would be expected in patients with OA [49]. In the same manner, systemic markers of cartilage turnover and diffuse synovitis similar to those present in OA, and not only in the impingement zone, have been identified in early-stage patients with FAI [49,50,51]. The presence of an arthritic process at the time of surgery should make us question the value of preservation surgery because of the poor results reported in joints with arthritic changes [16,52].

In this context, surgeons should focus on the real effect of the ostheochondroplasty on cartilage. The Australian FASHIoN RCT, published in 2021, reported some clarifying results about such effects. The data of semi-quantitative MRI analysis demonstrated worse cartilage and labral scores in the arthroscopic group at 12 months in comparison with the physiotherapy group. The dGEMRIC outcome was −59 ms (95% CI. 137.9 to 19.6) (*p* = 0.14), with the direction of the effect favoring the physiotherapy group in terms of hip cartilage metabolism [53]. The decline that the authors have seen in the dGEMRIC indices was found in prior investigations, and the magnitude of the decline found in our surgical group is broadly consistent with prior studies [54].

Our study is not free from limitations. The number of patients was not balanced 1:1:1, but this distribution reflects the proportion of patients who had surgical arthroplasty in our center. In addition, although true, the level of significance and the magnitude reported indicate that the statistical power was sufficient to be able to detect any differences between groups.

To reduce the intra- and interobserver differences in the measurement of the alpha angle [50], the radiological measurements were done by five different researchers.

## 5. Conclusions

Elevated acetabular angle, head extrusion and decreased Wiberg angle characterize hip osteoarthritis at younger ages and should be the focus of hip preservation surgery in terms of osteoarthritis prevention. Pincer-type FAI (higher Wiberg and lower Tönnis angles) and higher alpha angle (CAM) are correlated with the development of later OA. These results shed doubt on applying the hip preservation surgery concept to osteoarthritis prevention in FAI, especially in Pincer-type FAI patients.

## Figures and Tables

**Figure 1 diagnostics-12-01128-f001:**
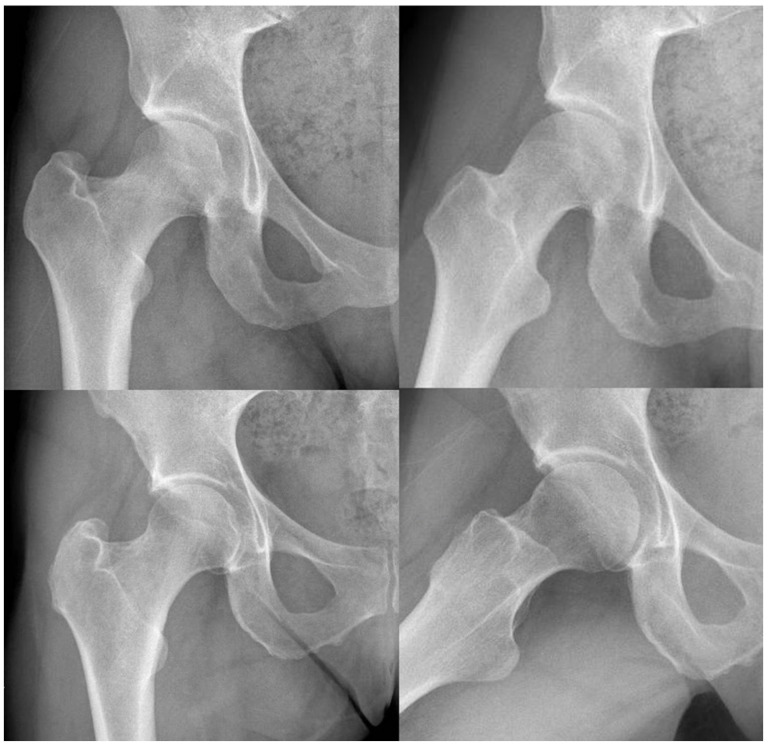
Radiographic comparations of CAM-type (first) and pincer-type (second) FAI.

**Figure 2 diagnostics-12-01128-f002:**
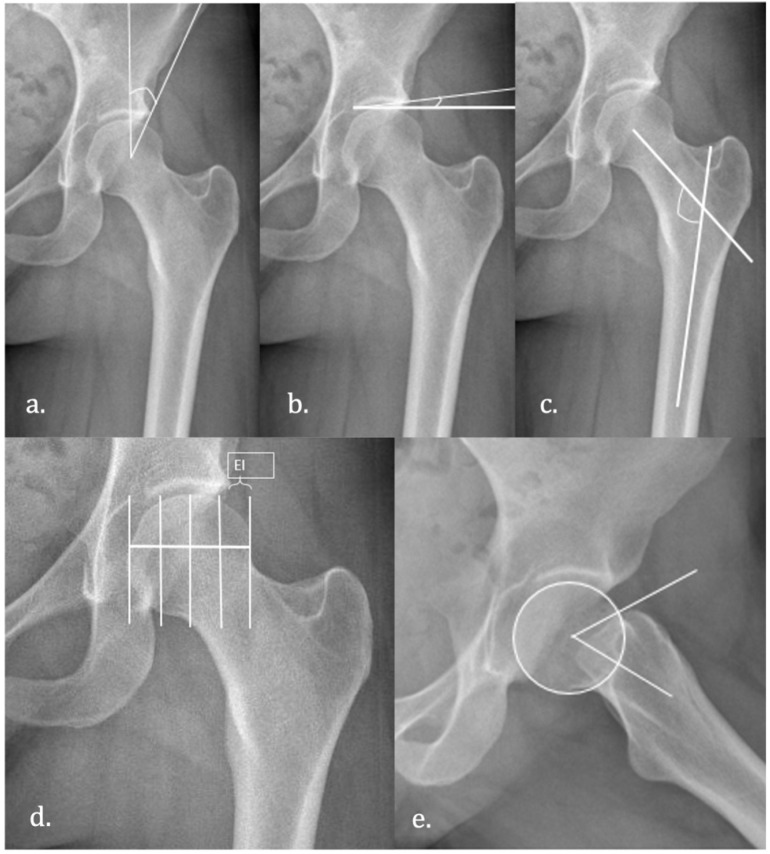
Hip angles. (**a**) Wiberg angle; (**b**) Tönnis angle; (**c**) centrum–collum–diaphyseal angle (**d**) extrusion index and (**e**) alpha angle.

**Table 1 diagnostics-12-01128-t001:** Demographic data (*n* = 400). Data are presented as mean (SD).

Age	64.75 (11.1)
Sex (M:F)	236:164
Size (R:L)	222:178
BMI	28.12 (4.42)
Wiberg angle (°)	38.2 (14.21)
Acetabular Angle (°)	10.86 (9.81)
Extrusion Index (%)	18.02 (12.07)
Alpha Angle (°)	68.59 (18.66)
Cervical–Diaphyseal angle (°)	131.45 (8.83)

**Table 2 diagnostics-12-01128-t002:** Group Demographic Data. Mean (SD). No differences in BMI between groups were assessed (*p* > 0.05).

	Group 1 (*n* = 147)	Group 2 (*n* = 155)	Group 3 (*n* = 98)
Sex (M:F)	98:49	87:68	50:48
Size (R:L)	84:63	84:71	54:44
BMI	27.65 (4.54)	28.77 (4.29)	27.83 (4.36)
Wiberg angle (°)	30.54 (14.63)	41.73 (11.20)	44.77 (12.02)
Tönnis angle (°)	15.35 (8.77)	8.94 (8.63)	7.11 (10.57)
Extrusion Index (%)	22.12 (4.54)	16.73 (11.47)	13.89(11.03)
Alpha angle (°)	61.78 (12.87)	71.34 (17.15)	73.64 (19.46)
Cervical–Diaphyseal angle (°)	133.17 (8.96)	130.60 (8.61)	130.28 (9.34)

**Table 3 diagnostics-12-01128-t003:** Angle comparisons between groups. Group 1, ≤60 years old; group 2, 61–74 years old; group 3, ≥75 years old.

	Group	Difference	IC (95%)	*p*
Wiberg angle (°)	≤60 vs. 61–74	−11.39	−15.036–(−7.75)	<0.01
	≤60 vs. ≥75	−14.43	−18.58–(−10.2)	<0.01
	60–74 vs. ≥75	−3.04	−6.69–0.61	0.133
Tönnis angle (°)	≤60 vs. 61–74	6.41	4.00–8.8	<0.01
	≤60 vs. ≥75	8.24	5.12–11.36	<0.01
	60–74 vs. >75	1.82	−1.23–4.9	0.395
Extrusion index (%)	≤60 vs. 61–74	5.39	2.12–8.66	<0.01
	≤60 vs. ≥75	8.23	4.61–11.85	<0.01
	61–74 vs. >75	2.83	−0.62–6.2	0.65
Alpha angle (°)	≤60 vs. 61–74	−9.55	−14.65–(−4.46)	<0.01
	≤60 vs. ≥75	−11.86	−18.00–(−5.71)	<0.01
	61–74 vs. ≥75	−2.3	−8.23–3.62	0.726
Cervical–Diaphyseal angle (°)	≤60 vs. 61–74	−1.11	−6.83–4.61	0.954
	≤60 vs. ≥75	−0.29	−6.75–6.16	0.999
	61–74 vs. ≥75	0.81	−3.90–5.53	0.966

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
