# Peer review of "Hip Preservation Surgery in Osteoarthritis Prevention: Potential Benefits of the Radiographic Angular Correction"

_diagnostics, 2022, doi:10.3390/diagnostics12051128_

Round 1

Reviewer 1 Report

I commend the authors for their research entitled "Hip Preservation Surgery in osteoarhtritis prevention: Potencial benefits of the radiographic angular correction". The manuscript is well written, easy to follow, and the conclusions are based on the results. I would suggest only some minor improvements: (1) Introduction - please spell out the "CAM"-type of FAI when first mentioned and elaborate the difference between pincer-type and CAM-type of FAI to be more understandable for the non-specialist readers (a typical radiograph of both entities would be much appreciated). (2) Materials and Methods - please use the standard term Centrum-Collum-Diaphyseal (CCD) angle instead of "Cervical-Diaphesal angle". (3) Figure 1 is not labelled correctly - there are no schemes. (4) Conclusions are an important part of the manuscript. Please elaborate, what is unique in your study and what important message you are bringing to the orthopaedic community. (5) Please check spelling: Page 8, line 224 ... or with arthroscopy saline serum.[4446]; Page 8, line 254 ... there is often poor intra-and inter-observer reliability agreement for the alpha angle.50 (6) Please check References' numbering and add reference number and date of IRB approval.

Author Response

We appreciate the comments raised by the reviewers that have undoubtedly contributed to enrich the manuscript. A point-by-point response to the comments is detailed below.

We hope this revised version meets now the standards of quality of Diagnostics.

Reviewer 1:

Question#1. Introduction - please spell out the "CAM"-type of FAI when first mentioned and elaborate the difference between pincer-type and CAM-type of FAI to be more understandable for the non-specialist readers (a typical radiograph of both entities would be much appreciated).

Response: The definitions of pincer-type and CAM-type of FAI have been added into the “Introduction” section as follows (page 1-2):

“There are three types of FAI. CAM-type is a hump in the transition area between the femoral head and the neck, so with the flex and rotate the hip, this hump hits the labrum and the acetabulum. Pincer-type is an overgrowth of the acetabular rim that causes friction against the head of the femur and its cartilage when flexing and rotating the hip. The third type is when CAM-type and Pincer-type deformity appear in the same patient.”

 We have added a new figure (Figure 1) with the radiographic differences of pincer-type and CAM-type.

 Question#2. Materials and Methods - please use the standard term Centrum-Collum-Diaphyseal (CCD) angle instead of "Cervical-Diaphesal angle".

Response: We have incorporated your suggestion in the manuscript. This has been added into the Material and Methods as follows:

The Centrum-Collum-Diaphyseal (CCD) angle is the angle formed by the anatomical axis of the femur and the axis of the femoral neck. An angle <125º is known as Coxa vara.

Question#3 Figure 1 is not labelled correctly - there are no schemes.

Response: We have incorporated your suggestion in the manuscript. This has been added into the Material and Methods as follows:

Figure 2. Hip angles: Hip angles a. Wiberg angle; b. Tönnis angle;                            c. Centrum-Collum-Diaphyseal angle d. Extrusion index and e. Alpha angle

Question#4 Conclusions are an important part of the manuscript. Please elaborate, what is unique in your study and what important message you are bringing to the orthopaedic community.

Response: We have incorporated your suggestion in the manuscript.

Elevated acetabular angle, head extrusion, and decreased Wiberg angle, characterize hip osteoarthritis at younger ages and should be the focus of hip preservation surgery in terms of Osteoarthritis Prevention. Pincer-type FAI (higher Wiberg, and lower Tönnis angle), and higher alpha angle (CAM) are correlated with the development of later OA. These results shed doubt on applying the hip preservation surgery concept in terms of Oste-oarthritis prevention in FAI, especially in Pincer-type FAI patients.

Question#5 Please check spelling: Page 8, line 224 ... or with arthroscopy saline serum.[44–46]; Page 8, line 254 ... there is often poor intra-and inter-observer reliability agreement for the alpha angle.50

Response: We have incorporated your suggestion in the manuscript.

In fact, the clinical results of osteochondroplasty associated with labral repair in several randomized clinical trials are not superior to a full program of physical therapy or arthroscopic saline lavage without performing any arthroscopic hip surgery.

To reduce the intra and interobserver difference in the measurement of the alpha angle [50], the radiological measurements were done by 5 different researchers.

Question#6 Please check References' numbering and add reference number and date of IRB approval.

Response: We have incorporated your suggestion in the manuscript.

Reviewer 2 Report

The authors aimed to describe the morphology associated with the development of OA in three different age groups. These data will contribute to define the morphology associated with early and late hip OA.

The study is easy to follow, but few issues should be improved. Some of the comments that would improve the overall quality of the study are:

  1. Authors must pay attention to the technical terms acronyms they used in the text. Please better stated the aim of the study in the abstract section.
  2. English language needs to be revised.
  3. Conclusion Section: This paragraph required a general revision to eliminate redundant sentences and to add some "take-home message".

Author Response

We appreciate the comments raised by the reviewers that have undoubtedly contributed to enrich the manuscript. A point-by-point response to the comments is detailed below.

We hope this revised version meets now the standards of quality of Diagnostics.

Question#1. Authors must pay attention to the technical terms acronyms they used in the text. Please better stated the aim of the study in the abstract section.

Response: We have incorporated your suggestion in the manuscript. This has been added into the Abstract as follows:

The aim of the study is to establish the correlation between the anatomical angles of the hip (be-fore and after surgery for femoracetabular impingement or adult hip dysplasia) with the development of early osteoarthritis.

Question#2. English language needs to be revised.

Response: A native-English language reviewer from the University of Navarra has reviewed the English of the manuscript.

Question#3. Conclusion Section: This paragraph required a general revision to eliminate redundant sentences and to add some "take-home message".

Response: We have incorporated your suggestion in the manuscript.

Elevated acetabular angle, head extrusion, and decreased Wiberg angle, characterize hip osteoarthritis at younger ages and should be the focus of hip preservation surgery in terms of Osteoarthritis Prevention. Pincer-type FAI (higher Wiberg, and lower Tönnis angle), and higher alpha angle (CAM) are correlated with the development of later OA. These results shed doubt on applying the hip preservation surgery concept in terms of Oste-oarthritis prevention in FAI, especially in Pincer-type FAI patients.